# Generating Fundus Fluorescence Angiography Images from Structure Fundus Images Using Generative Adversarial Networks

**Wanyue Li**[1,2,3]                                                        WANYUELI93@126.COM
**Wen Kong**[1,2,3]                                                        KONGWEN_WORK@163.COM
[1] *University of Science and Technology of China, Hefei, 230041, China.*
[2] *Jiangsu Key Laboratory of Medical Optics, Suzhou, 215163, China.*
[3] *Suzhou Institute of Biomedical Engineering and Technology, Chinese Academy of Sciences, Suzhou, 215163, China.*

**Yiwei Chen**[2,3]                                                        YIWEI.CHEN@SIBET.AC.CN
**Jing Wang**[1,2,3]                                                       CRYSKING_WJ@163.COM
**Yi He**[2,3]                                                             HEYI_JOB@126.COM
**Guohua Shi**[*1,2,3,4]                                                   GHSHI_LAB@126.COM
[4] *Center for Excellence in Brain Science and Intelligence Technology, Chinese Academy of Sciences, Shanghai, 200031, China.*

**Guohua Deng**[5]                                                        CZDGH1975@SINA.COM
[5] *Department of Ophthalmology, the Third People's Hospital of Changzhou, Changzhou, 213000, China.*

## Abstract

Fluorescein angiography can provide a map of retinal vascular structure and function, which is commonly used in ophthalmology diagnosis, however, this imaging modality may pose risks of harm to the patients. To help physicians reduce the potential risks of diagnosis, an image translation method is adopted. In this work, we proposed a conditional generative adversarial network (GAN)-based method to directly learn the mapping relationship between structure fundus images and fundus fluorescence angiography (FFA) images. Moreover, local saliency maps, which define each pixel's importance, are used to define a novel saliency loss in the GAN cost function. This facilitates more accurate learning of small-vessel and fluorescein leakage features. The proposed method was validated on our dataset and the publicly available Isfahan MISP dataset with the metrics of peak signal-to-noise ratio (PSNR) and structural similarity (SSIM). The experimental results indicate that the proposed method can accurately generate both retinal vascular and fluorescein leakage structures, which has great practical significance for clinical diagnosis and analysis.

**Keywords:** Fundus Fluorescence Angiography Image, Structure Fundus Image, Image Translation, Generative Adversarial Network, Local Saliency Map.

## 1. Introduction

Data from the World Health Organization shows that more than 2.2 billion people have a vision impairment or blindness so far (Organization et al., 2019). And an imaging technique that can accurately reflect the microscopic fundus lesions is important for the diagnosis,

---

[*] Corresponding author

treatment, and prevention of ophthalmic diseases. Fluorescein angiography (FA) can reflect the damaged state of the retinal barrier in the eyes of living human, especially in the early stage of diseases, and dynamically capture the physiological and pathological conditions from the large vessels of the retina to the capillaries, which is regarded as the "gold standard" of retinal disease diagnosis (O'TOOLE, 2007). Despite the diagnostic benefits, physicians are increasingly reluctant to use angiography imaging technology because of its potential serious adverse effects (Yannuzzi et al., 1986; Musa et al., 2006; Schiffers et al., 2018). Although fundus structure imaging and fundus fluorescein angiography (FFA) are significantly different in imaging mode and image appearance, they both have many common features, such as vessels or granular structures. Therefore, using image generation method to generate the corresponding FFA image from the fundus structure image can help physicians to diagnose with smaller potential risks in patients and relatively reduce the need for actual angiographic imaging.

FFA image generation can be defined as an image-to-image translation problem; that is translating the fundus image from the structure domain to FA domain. The idea of image-to-image translation can go back to Hertzmann's "Image Analogies" work (Hertzmann et al., 2001), who employ a non-parametric texture model (Efros and Leung, 1999) on a single input-output training image pair. In recent years, deep learning and generative adversarial networks (GANs) have become popular approaches for image synthesis and translation, and has been utilized for various medical imaging modalities (Nie et al., 2017; Jin et al., 2019; Wolterink et al., 2017; Lee et al., 2019; Kida et al., 2019). According to the characteristics of datasets, algorithms that tackle the image synthesis and translation problem can be mainly divided into paired or unpaired methods. Unpaired methods are mainly used to solve the problem that paired images are difficult to obtain. Since the images that are taken from same patients at same anatomical locations are not easy to acquire, the unpaired methods are widely used in medical image synthesis and translation tasks, including synthesizing CT images from MR images (Nie et al., 2017; Jin et al., 2019; Wolterink et al., 2017), planning CT image from Cone-beam CT image (Costa et al., 2017), etc. In terms of retina image synthesis, Schiffers et al. (Schiffers et al., 2018) proposed a method based on CycleGAN (Zhu et al., 2017) to generate FFA images from retinal fundus images. In paper (Li et al., 2019), the authors also proposed an unsupervised FFA image synthesis method via disentangled representation learning based on unpaired data. Both of the above methods demonstrate that the unpaired methods can achieve the translation from fundus image domain to FFA image domain, however, the FFA image generated by this kind of method cannot be directly used in disease diagnosis. For medical image generation tasks, it is crucial to precisely generate the position and morphology of lesions. The unpaired methods only learn the global features from one domain to another domain, and cannot generate accurate pathological structures. The drawbacks of the unpaired methods applied to FFA image synthesis task has also been illustrated in the experimental comparison in (Hervella et al., 2019).

Paired image translation methods can directly learn the mapping relationship from input to output images, which could have better performance in learning the detailed information of images. This kind of method also has been applied to FFA image synthesis. Hervella et al. (Hervella et al., 2018) constructed a Unet architecture with fundus image as input and FFA images as output to learn a direct mapping between two domains. However, without

the help of adversarial learning, their model can only learn a pixel-to-pixel mapping, which results in an unsatisfied performance. Moreover, due to the scarcity of paired data, the model would easily become overfitting, which deteriorates the generalization ability of the model. In work (Hervella et al., 2019), the authors utilized a generator composed of an encoder, 9 residual blocks, and a decoder to synthesize FFA image based on the paired data. Although this method has better performance than the Unet and unpaired methods, there is also considerable room for improving in the reconstruction of the pathological structures. Furthermore, their model was trained and tested on totally 118 retinography-angiography pairs, such a small amount of data may affect the credibility and generalization ability of the model to some extent.

As noted above, the existing FFA image generation methods mainly have the following two problems. (1) None of the existing methods can accurately generate the pathological structures, which seriously affects the practical clinic application of these methods. (2) The quantity and variety of the training and testing data are insufficient, which may be the main cause of the unsatisfied results and the poor generalization ability. In addition, due to the limitations of data, researchers cannot select data according to the characteristics of fundus angiography and clinical demands, which unable to ensure the medical significance of their findings. To solve the first problem, we proposed an FFA image generation method based on conditional GAN that has some novelties in the design of the loss function. In this approach, the loss function was formulated as a combination of the global loss and local loss, where the local loss is based on the local saliency map of the label FFA image. Using this local loss can ensure the accurate generation of the pathological structures in the synthesis FFA image. Moreover, the discriminator was designed based on the idea of PatchGAN (Li and Wand, 2016; Isola et al., 2017) model, which penalizes the image structure at the scale of patches instead of the whole images, and can help reconstruct image details and improve the training efficiency. For the second problem, the data we used for training and testing were all collected from the hospital; and according to the characteristic that FFA is the only method to detect leakage, we choose the generation of FFA images with fluorescein leakage as the focus of this study. There are three common types of leakage that can be observed in fluorescein leakage: optic disc, large focal, and punctate focal leakage (Figure 1(b)-(c)). These kinds of leakage have two main characteristics. One of the characteristics is that the leakage of early angiography usually does not appear or is not obvious, but its size and brightness will increase in the late stage. The other one is that the lesion with leakage cannot be observed on the fundus structure image in the early stage of the disease, and can only be determined by using FFA. Therefore, based on the characteristics of FFA and fluorescein leakage, the main study in this paper is to generate the corresponding late FFA images from the fundus structure images, so as to achieve the auxiliary diagnosis of the diseases with such fluorescein leakages. The rest of this paper is organized as follows: in section 2, the datasets and the proposed method are discussed. The quantitative and qualitative comparison of the results are described in section 3. In section 4, we summarize our work and discuss the experimental results.

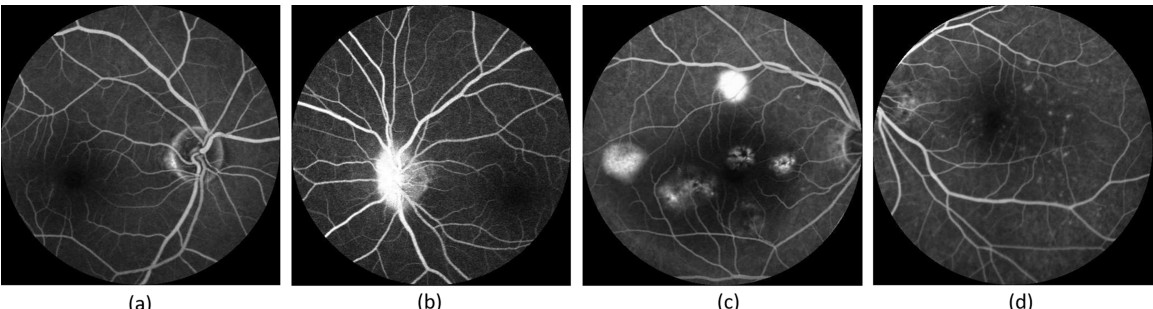

Figure 1: Illustration of the types of FFA image focused on this study. (a) Normal FFA image. FFA image with (b) optic disc leakage; (c) large focal leakage; (d) punctate focal leakage.

## 2. Method

### 2.1. Datasets

The dataset used in this study were taken with a Spectralis HRA (Heidelberg Engineering, Heidelberg, Germany) between March 2011 and September 2019 at the Third People's Hosptital of Changzhou (Jiangsu, China), which is called "HRA dataset" here. The image types in our "HRA dataset" including normal FFA images and the FFA images with optic disc, large focal, and punctate focal leakage, as shown in Figure 1. Since the late angiography images are more helpful for disease diagnosis, we select the FFA images captured between 5 and 6 min as the label FFA images. This dataset initially contained 1,450 retinography-angiography image pairs from 1,450 eyes of 802 patients (326 female, 476 male, ranging in age from 7 to 86 years), and 1 picture per eye. The field of views of these images including 30°, 45°, 60°, and the resolution of each image is $768 \times 768$ pixels.

The proposed method is a paired method which needs paired images as the dataset. However, the collected retinography-angiography image pairs always unaligned, then the datasets preprocessing is needed. The main steps of datasets preprocessing as follows.

**Multimodal registration:** The registration of the retinography-angiography pairs is needed for building a pixel-wise correspondence. The multimodal registration is performed using the method proposed in (Wang et al., 2015). For some low-quality images, this method may not work; thus, we manually delete the image pairs that are aligned inaccurately.

**Obtain aligned image pairs:** The aligned retinography-angiography images were obtained by extracting 512x512 patches from the aligned part in the image pairs. Eye fundus photographies in our dataset displayed in two forms, one is displayed as a square image(Figure 3), and the other one displays the retina in a circular region of interest (ROI) (Figure 1). For the former, the aligned images pairs can be obtained by extracting the corresponding patches from the aligned part in both structure and FFA images. For the latter, the extracted patches in the aligned part may contain the irregular black areas, this will affect the learning of the image displayed forms and even the generation of important information. To ensure the generalization ability of the model in different displayed forms,

we first designed a mask with a circle whose diameter is the width of the image patch, and the pixel values are 1 in or on the circle and that are 0 out of the circle, Then, the extracted image patches were multiplied with the mask, we can finally obtain the aligned image pairs with circular ROI.

Through the registration operation, our aligned "HRA dataset" finally contained 1,230 successfully aligned retinography-angiography image pairs from 1,230 eyes of 727 patients (297 female, 430 male, ranging in age from 7 to 86 years), of which normal vessels, optic disc leakage, large focal leakage, and punctate focal leakage image pairs are 302, 151, 448, and 329, respectively. Then, 20% of the image pairs were randomly selected from each of the data types as the testing set, the remaining 80% were used as the training set. Finally, after the aligned patch extracting operation, we totally have 10,636 image patches with $512 \times 512$ pixels used for training, and 2,699 image patches used for testing.

### 2.2. Local saliency map calculating

Local saliency map of the FFA image is regard as the restricted condition to ensure the accurate generation of fluorescein leakage structures. Most of the commonly used methods (Zhao et al., 2017, 2015) to generate saliency map of FFA image are pixel-based methods, which are time-consuming and thus do not meet the requirements of our work. In this section, we proposed a simple, effective, and fast method for FFA saliency map generation.

An observed fundus image $I$ can be modeled as the combination of background image $I_b$ and the foreground image $I_f$, where $I_b$ is the idea image of a retinal fundus free of any vascular structure, optic disc, or visible lesion; and $I_f$ is the vascular structure, optic disc, or visible lesion which are the most attentive parts in the image generation and detection tasks. Since the vascular structure, optic disc, and fluorescein leakage in FFA image are all highlighted, extracting the foreground of the FFA image can help us obtain its saliency map.

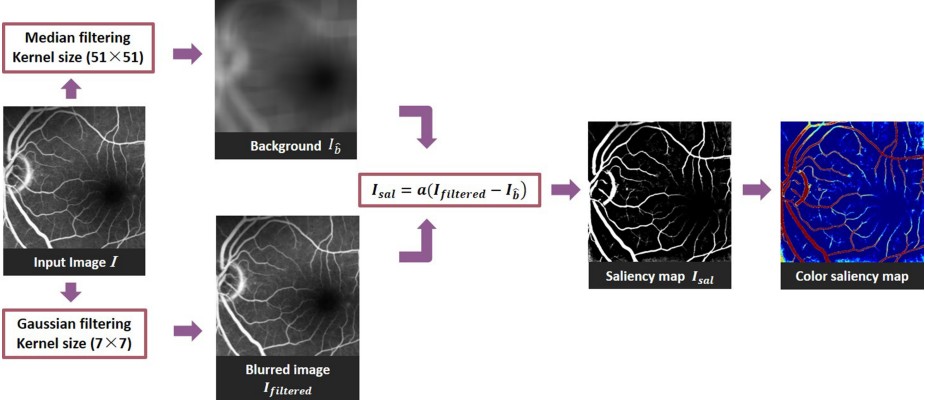

Figure 2: Calculating process of FFA saliency map.

The calculating process of FFA saliency map is illustarted in Figure 2. Firstly, to estimate the background of FFA image, a median filtering with a mask of size $51 \times 51$ is performed, which is large with respect to the maximal vessel diameter (about 15 pixels),

but small with respect to the optic disc (about 120 pixels). Secondly, a Gaussian filtering with a mask of size $7 \times 7$ is used for the denoising of the input image. Finally, the saliency map can be obtained by subtracting the filtered image and the background image, and can be expressed as:

$$I_{sal} = a(I_{filtered} - I_{\hat{b}}),\qquad(1)$$

where $I_{sal}$ is the saliency map; $I_{filtered}$ is the original FFA image filtered by a Gaussian filter; $I_{\hat{b}}$ is the estimated background image; $a$ is a parameter factor which determine the contrast of image.

The calculated saliency maps and corresponding FFA images are presented in Figure 3, the warmer color indicates the more saliency regions, and the cooler color shows the less saliency regions, and the appearance of the vessels, optic disc, and leakage are highlighted compared to the original image.

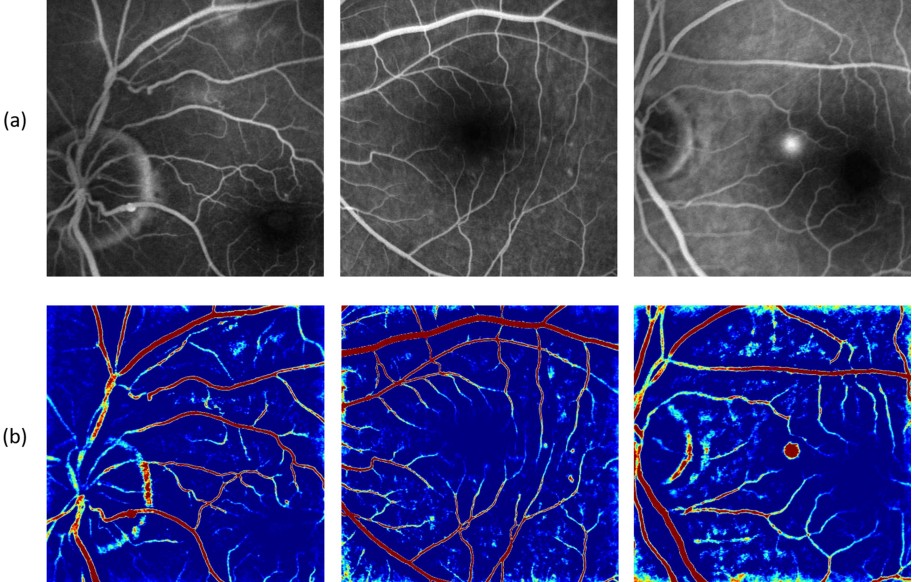

Figure 3: Examples of generated saliency maps of FFA images. (a) Original FFA image; (b) Corresponding saliency map.

### 2.3. Conditional adversarial network for FFA generation

In this section, we introduce our conditional GAN architecture for translating the structure fundus images to FFA images. As illustrated in Figure 4, the proposed network is mainly composed of two important parts (generator G and discriminator D). The generator (left part of Figure 4) based on a series of residual blocks, and its primary mission is to generate a FFA image from an input structure fundus image. The discriminator (right part of Figure 4) serves to distinguish the synthetic FFA image from the corresponding real FFA image, which can also be viewed as a guide for the training of generator.

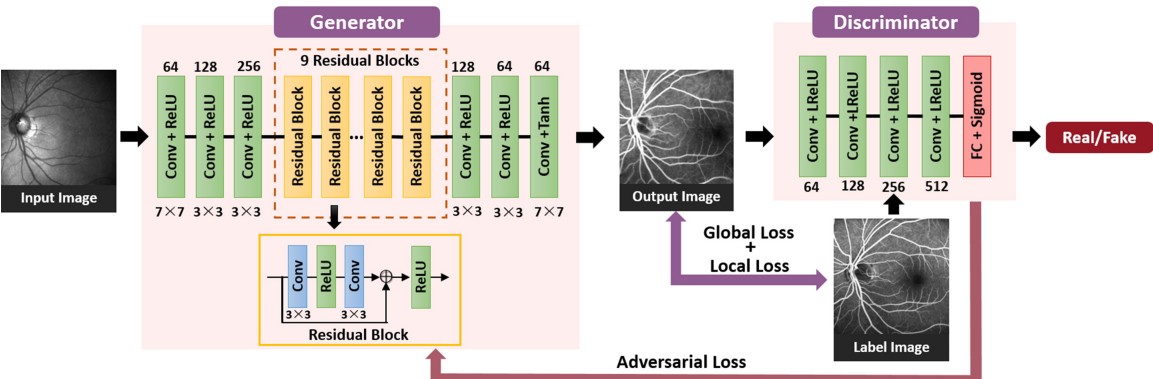

Figure 4: Proposed model architecture. Our model is composed of a Resnet-based generator and a PatchGAN-based discriminator. (conv: convolutional layer; ReLU: rectified linear unit; Tanh: TanHyperbolic function; LReLU: leaky rectified linear unit; FC: fully connected layer)

The main purpose of the proposed method is to learn a mapping relationship from an input structure fundus image $I_S$ and a random noise vector $n$ to generate a synthesis FFA image $I_F$ by solve the following min-max problem:

$$\min_G \max_D E_{I_S, I_F \sim p(I_S, I_F)}[logD(I_S, I_F)] + E_{I_S \sim p_{(I_S)}, n \sim p(n)}[log(1 - D(I_S, G(I_S, n)))]. \quad (2)$$

The architecture of the proposed generator and discriminator networks are shown in Figure 4. As illustrated in the left part of Figure 4, the core of the proposed generator is a series of residual blocks, each composed of two convolutional layers with kernel size $3 \times 3$, and each convolutional layer is followed by a rectified linear unit (ReLU) activation function. The proposed discriminator is inspired by the idea behind PatchGAN, which has shown good performance in image details learning in many image generation tasks (Isola et al., 2017; Kupyn et al., 2018; Li et al., 2020). As shown in the right part of Figure 4, the proposed discriminator network is composed of five convolutional layers with kernel size $4 \times 4$, and the first four convolutional layers are followed by a leaky ReLU (LReLU) activation function. The last convolutional layer is used for mapping to a one-mapping output, followed by a sigmoid function. Using this network as a discriminator can help generating image details and improving the training efficiency.

## 2.4. Loss Function

To ensure that the generated FFA image resembles the label image and accurately generate the fluorescein leakage, we formulate the loss function as a combination of global and local loss, where global loss is a combination of adversarial, pixel-space, and feature-space loss, whose main function is to ensure the reconstruction of the whole information of the image; and local loss is based on the local saliency loss, that is used to make the network concentrate

more on the vascular and leakage structures reconstruction. The proposed loss function is defined as follows:

$$L = L_{Global} + L_{Local} = (L_{GAN} + \alpha L_{pixel} + \beta L_{perceptual}) + \gamma L_{sal}, \tag{3}$$

where $\alpha$, $\beta$, and $\gamma$ are the experimentally determined hyperparameters that control the effect of pixel-space, feature-space, and local saliency loss. Pursuing the balance among pixel-space, feature-space, and local saliency loss is not a trivial task. Given over-weight to feature-space loss will result in the loss of image intensity information after reconstruction (Kupyn et al., 2018), and give a low-weight to local saliency loss will make the network cannot pay enough attention to local information; thus, feature-space loss should have a lower weight than pixel-space loss, and local saliency loss should have a relatively high weight. After the massive experiments, we found that $\alpha = 100$, $\beta = 0.001$, and $\gamma = 1$ are the suitable parameters to achieve better performance.

**Adversarial loss:** The generative loss over all training samples is defined as follows:

$$L_{GAN} = \sum_{n=1}^{N} -log D_{\theta_G}(G_{\theta_G}(I_S)). \tag{4}$$

**Pixel-space loss:** Similarly to pix2pix model (Isola et al., 2017), we also adopt an L1 loss to encourage the generated synthesis FFA image $G_{\theta_G}(I_S)$ from a structure fundus map $I_S$ to be close to the label FFA image $I_F$. The L1 loss is calculated as follows:

$$L_{pixel} = \frac{1}{WH}\sum_{x=1}^{W}\sum_{y=1}^{H}|(I_F)_{x,y} - (G_{\theta_G}(I_S))_{x,y}|. \tag{5}$$

**Feature-space loss:** Perceptual loss is used as the feature-space loss to help reconstruct textural information. Perceptual loss is based on the difference of generated and label image CNN feature maps, which can help generate the images with high perceptual quality. It can be given by:

$$L_{perceptual} = \frac{1}{W_{i,j}H_{i,j}}\sum_{x=1}^{W_{i,j}}\sum_{y=1}^{H_{i,j}}(\phi_{i,j}(I_F)_{x,y} - \phi_{i,j}(G_{\theta_G}(I_S))_{x,y})^2, \tag{6}$$

where $\phi_{i,j}$ is the feature map obtained by the $jth$ convolution before the $ith$ maxpooling layer within the VGG19 network, pretrained on ImageNet (Simonyan and Zisserman, 2014), and $W_{i,j}$ and $H_{i,j}$ are the dimensions of $\phi$.

**Local Saliency loss:** To ensure the generated FFA image with accurate fluorescein leakage structures, we proposed a novel saliency loss which measures the difference between the local saliency maps of $G_{\theta_G}(I_S)$ and $I_F$ by comparing their local landmarks. It can be expressed as:

$$L_{sal} = \frac{1}{W_{i,j}Hi,j}\sum_{x=1}^{W_{i,j}}\sum_{y=1}^{H_{i,j}}((I_F^{sal})_{x,y} - ((G_{\theta_G}(I_S))_{x,y}^{sal})^2, \tag{7}$$

where $I_F^{sal}$ and $(G_{\theta_G}(I_S))^{sal}$ denote the local saliency maps of $I_F$ and $G_{\theta_G}(I_S)$.

## 3. Experimental results

To demonstrate the effectiveness of the proposed method in FFA image synthesis, we conducted a series of experiments on both our HRA dataset and the publicly available Isfahan MISP dataset (Hajeb Mohammad Alipour et al., 2012). Compared with other state-of-the-art methods also show the superiority of the proposed method.

All experiments were carried out in an Ubuntu 16.04 + python 3.6 environment. We trained the proposed model for 200 epochs using the Adam optimizer with momentum parameters $\beta_1 = 0.5$, $\beta_2 = 0.999$. The initial learning rate was set to 0.0002 for both generator and discriminator. After the first 100 epochs, the learning rate was linearly decayed to zero over the next 100 epochs. The proposed model was trained with a batch size equal to 1, which showed empirically better result on validation. It takes about 70 hours on two GeForce GTX 1080Ti GPUs to train our HRA dataset, and nearly 3 hours to train the Isfahan MISP dataset.

### 3.1. Evaluation metrics for aligned image patches

The testing results were evaluated with the criteria of peak signal-to-noise ratio (PSNR) and structural similarity (SSIM). These two metrics are commonly used for quantitatively evaluating image reconstruction quality, and are defined as follows:

$$MSE = \frac{\sum_{n=1}^{N}(x^n - y^n)}{N}, \tag{8}$$

$$PSNR = 10 \times lg(\frac{255^2}{MSE}), \tag{9}$$

$$SSIM(x,y) = \frac{(2\mu_x\mu_y + c_1)(2\sigma_{xy} + c_2)}{(\mu_x^2 + \mu_y^2 + c_1)(\sigma_x^2 + \sigma_y^2 + c_2)}, \tag{10}$$

where $N$ is the size of image; $x^n$ and $y^n$ are the $nth$ pixels of original image $x$ and processed image $y$; $\mu_x$ and $\mu_y$ are the averages of $x$ and $y$; $\sigma_x^2$ and $\sigma_y^2$ are the variance of $x$ and $y$; and $\sigma_{xy}$ is the covariance of $x$ and $y$.

### 3.2. Results comparison on HRA dataset

To demonstrate the effectiveness of the proposed method, we compared the results of the proposed method with those of two state-of-the-art methods. One of the methods is Cycle-GAN (Zhu et al., 2017), which is one of the most popular unsupervised image translation methods, and has been applied to FFA image synthesis task in (Schiffers et al., 2018). The other one is pix2pix (Isola et al., 2017) method which is widely used in medical image translation tasks. For fair comparison, these two models were both trained and tested on HRA dataset. Since the introduction of PatchGAN and local saliency loss makes our network has good performance on the detailed information and the vascular and leakage structure generation, we also compared the final results of the proposed method with the network trained without local saliency loss or PatchGAN. The configuration of regular discriminator mainly refers to the setting of the discriminator with receptive field size of 256 x 256 in the "pix2pix" work (Isola et al., 2017).

Figure 5 illustrates the synthesis results of four types of FFA images taken from the test set. Each column of Figure 5 shows from the left to right the input structure fundus image, the real FFA image, and the image generated by CycleGAN, pix2pix, the network without local saliency loss, the network with regular discriminator, and the proposed method. And the first to the fourth row of Figure 5 is the synthesis result examples of normal FFA image and the image with optic disc leakage, large focal leakage, and punctate focal leakage, respectively. For better comparison, we zoom in on the local regions with yellow boxes.

As shown in Figure 5(c1)-(c4), the images generated by CycleGAN can roughly reflect the characteristics of the FFA image; however, this method only externally renders the structure fundus image according to the extracted FFA features to make it look like an FFA image; but cannot accurately generate the leakage or even some retinal vessel structures. As described in Figure 5(d1)-(d4), pix2pix method can generate images with more detailed information; however, the images generated by pix2pix could contain some pseudo information, such as small vessels (Figure 5(d1)) and leakages (red circle in Figure 5(d3)) that should not exist. Furthermore, this method also cannot accurately learn the location or structure information of leakages (Figure 5(d3) and (d4)). As illustrated in Figure 5(e1)-(e4) and (f1)-(f4), the FFA images generated by the resnet-generator with pixel and perceptual loss have better performance and detailed information, but also contain some pseudo vessels (green circle in Figure 5(e1) and e(2)) and cannot generate accurate leakages. And the images generated by the regular discriminator also have the problem that cannot generate leakages accurately but generate some pseudo structures. As shown in Figure 5(g1)-(g4), the images generated by the proposed method have more accurate detailed information, and can accurately generate the small vessels. Although the proposed method performs not very well in the leakage details generation (Figure 5(g3) and (g4)), it can accurately generate the location and rough structure of leakages, which also indicates the superiority of the proposed method.

For quantitative comparison, the synthetic FA images translated by CycleGAN, pix2pix, the proposed network without local saliency loss, the network without PatchGAN, and the proposed method were all evaluated with the metrics of PSNR and SSIM. As seen in Table 1, the proposed method significantly outperforms the other four methods. The average PSNR of the proposed method is 4.83 dB, 1.30 dB, 0.66dB, and 1.37dB higher than that of the CycleGAN, pix2pix, the proposed network trained without local saliency loss, and the network without PatchGAN, respectively; and the average SSIM is 0.2706, 0.0830, 0.0768, and 0.1023 higher, respectively.

Table 1: Performance comparison with different methods tested on HRA dataset

| Metrics | CycleGAN | pix2pix | without $L_{sal}$ | without PatchGAN | the proposed method |
|---------|----------|---------|-------------------|------------------|---------------------|
| PSNR(dB) | 15.15 | 18.68 | 19.32 | 18.61 | **19.98** |
| SSIM | 0.5949 | 0.7825 | 0.7887 | 0.7632 | **0.8655** |

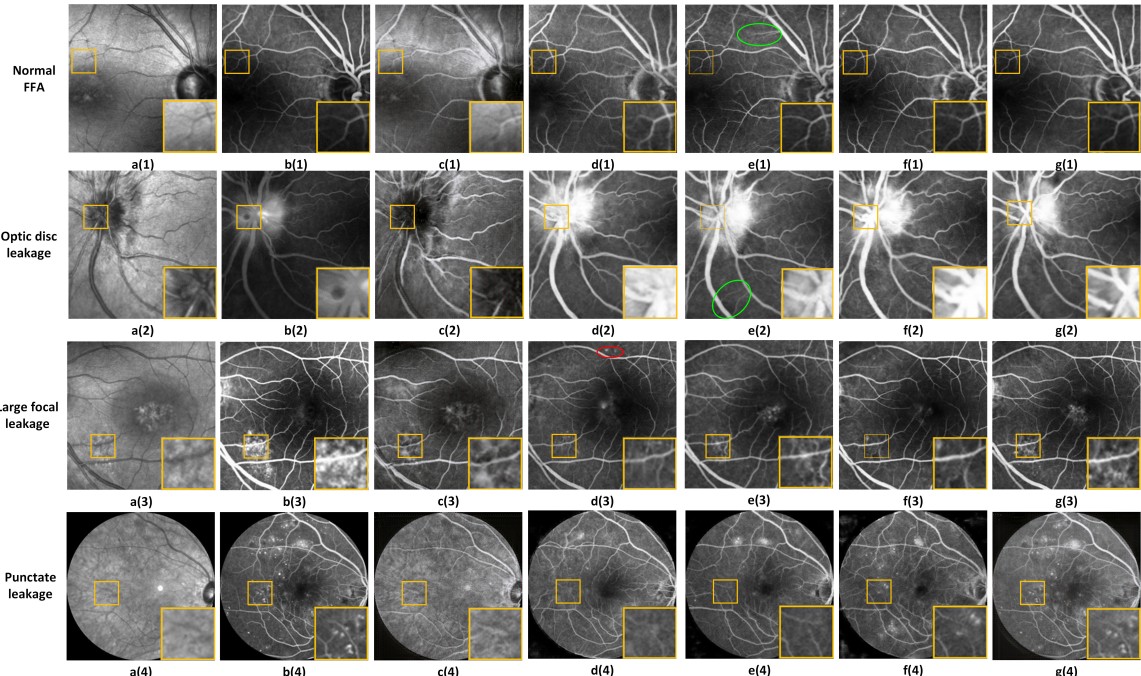

Figure 5: Synthesis results of four types of FFA images. (a1)-(a4) Structure fundus images; (b1)-(b4) Real FFA images; Generated FFA images from (c1)-(c4) the CycleGAN method; (d1)-(d4) the pix2pix method; (e1)-(e4) the proposed network without local saliency loss; (f1)-(f4) the proposed network without PatchGAN (with regular discriminator); and (g1)-(g4) the proposed method.

### 3.3. Results comparison on Isfahan MISP dataset

To further evaluate the effectiveness and universality of the proposed method, the proposed method was also trained and tested on the Isfahan MISP dataset, and was compared quantitatively and qualitatively with the four methods mentioned in section 3.2.

The publicly available Isfahan MISP dataset contains 59 retinography-angiography pairs divided in healthy and pathological cases. The images have a resolution of $720 \times 576$ pixels, and the image pairs in this dataset were also preprocessed by using the dataset preprocessed method proposed in section 2.1 and extracted as the patches with $360 \times 288$ pixels.

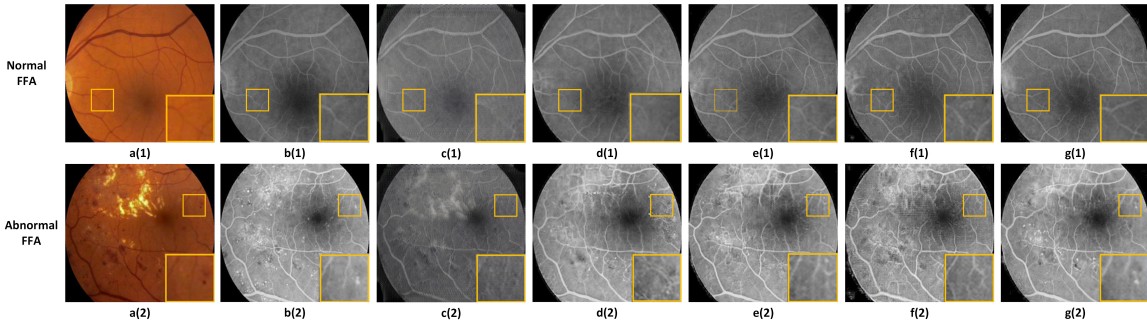

Figure 6: Synthesis results of normal and abnormal FFA images. (a1) and (a2) Structure fundus images; (b1) and (b2) Real FFA images; Generated FFA images from (c1) and (c2) the CycleGAN method; (d1) and (d2) the pix2pix method; (e1) and (e4) the proposed network without local saliency loss; (f1) and (f4) the proposed network without PatchGAN (with regular discriminator); and (g1) and (g2) the proposed method.

Figure 6 shows the examples of normal and abnormal images synthesis results; and the local regions are also zoomed in on with yellow boxes for better comparison. As illustrated in Figure 6(a1)-(g1), compared with the other four methods (Figure 6(c1)-(f1)), the proposed method (Figure 6(g1)) performs well in small vessels generation, and can generate more similar vascular structures to the real FFA image. As shown in Figure 6(a2)-(g2),the proposed method (Figure 6(g2)) also have better performance in leakages generation when compared with the other four cases (Figure 6(c2)-(f2)). This demonstrates the effectiveness of the proposed method in FFA image generation.

The tested results on Isfahan MISP dataset were also quantitatively compared with the metrics of PSNR and SSIM. As shown in Table 2, the proposed method surpassed the CycleGAN, pix2pix, the proposed network trained without local saliency map, and the proposed network trained without PatchGAN 5.51dB, 1.73dB, 0.17dB, and 1.42dB in PSNR, respectively; and 0.2469, 0.0830, 0.0600, and 0.0797 in SSIM, respectively. This implies that the proposed method has better performance on the FFA image generation task.

Table 2: Performance comparison with different methods tested on Isfahan MISP dataset

| Metrics | CycleGAN | pix2pix | without $L_{sal}$ | without PatchGAN | **the proposed method** |
|---------|----------|---------|-------------------|------------------|-------------------------|
| PSNR(dB) | 19.65 | 23.43 | 24.99 | 23.74 | **25.16** |
| SSIM | 0.5799 | 0.7438 | 0.7668 | 0.7471 | **0.8268** |

## 4. Discussion and Conclusion

Fluorescein angiography (FA) is a type of imaging commonly used in ophthalmology clinics that provides a map of retinal vascular structure and function by highlighted blockage of, and leakage from, retinal vessels. However, the potential adverse effects of FA imaging limit its usage to some extent. With the development of image synthesis and translation methods, some researchers also provide an idea that whether we can synthesize an FFA image from a fundus image to help reduce the potential diagnosis risks; and many works (Costa et al., 2017; Schiffers et al., 2018; Li et al., 2019; Hervella et al., 2019) about that have been presented. However, the presented methods all have two problems, one is that these methods cannot accurately generate the pathological structures, the other one is that the quantity and variety of the training and testing data are insufficient, which may effect the generalization ability of the model and lack of medical significance. In this paper, we proposed a conditional GAN-based FFA image generation method with a novel saliency loss function to ensure the accurate generation of the pathological structures in the synthesis FFA image.

According to the characteristics of FFA and fluorescein leakage, we choose the generation of FFA images with three common types of fluorescein leakage (large focal, puncate focal, and optic disc leakage) as well as normal FFA images as the focus of this study. As shown in Figure 5 and Figure 6, the proposed method has a good overall performance on the generation of normal FFA image and the image with leakages, but it performs unsatisfied on the leakage details generation. Therefore, improving the performance of leakage details generation will be the main task in our future work.

In this study, we use the PSNR and SSIM to evaluate the quality of the generated FFA image and the similarity between the generated and real FFA image. This evaluation method can illustrate the potential value of the FFA synthetic methods in medical applications, but it cannot effectively show that the generated FFA image is reliable and valuable enough for physicians. Unfortunately, we do not yet have a better and more acceptable approach to verify the reliability and medical value of the synthetic FFA images. But to find suitable and reliable measurement methods is the problem we have been thinking about and studying, maybe it will be solved in future work.

In addition, in this work, we split the training and testing sets at the image level not the participant level. Although this split operation may make a single person have images in both the training and testing sets, the pathological feature of the two eyes of the same person is not same. Therefore, splitting the dataset at the image level cannot affect the generalization ability and credibility of the proposed model.

To conclude, the proposed method has better performance in the generation of retinal vascular and fluorescein leakage structures when compared with other methods, which has great practical significance for clinical diagnosis and analysis.

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
