# OpenReview forum: "Generating Fundus Fluorescence Angiography Images from Structure Fundus Images Using Generative Adversarial Networks"
_MIDL.io/2020/Conference — MIDL 2020_

### Official Review · AnonReviewer1 · 2020-03-12
**Simple idea, good performance but lack of sufficient experiments.**

**Rating:** 3
**Confidence:** 3
**Recommendation:** Poster

**Summary:**

This paper have two major contributions. One is that local saliency map has been used in GAN loss, the other is the newly introduced loss, a combination of  global and local losses. The proposed method outperforms the other two comparison methods. However, the comparison experiments are not enough to prove their claims.


**Strengths:**

1. Main main strength of this work is the introduction of local saliency map, putting more weights on high-frequency regions. This strategy can efficiently improve the synthesis performance on details.
2. Introduction of the proposed method is very clear.
3. Paper is well written, easy to follow.

**Weaknesses:**

1. Since the network architecture is based on (inspired by) previous work (PatchGAN), I think the authors should add experiments to validate the performance with this baseline.
2. Ablation study should be added to validate the effectiveness of the proposed loss function.




**Justification Of Rating:**

The proposed method has achieved quite good performance compared to related works. The idea is simple and effective. The paper is well written. Hope the authors can add more comparison experiments to further prove their claims.

**Paper Type:**

methodological development

**Questions To Address In The Rebuttal:**

1. As the authors claim that "Most common used saliency maps are pixel-based methods,which are time-consuming and do not meet the requirements of our work", What do authors mean "do not meet the requirements"? Obviously, for training process, local saliency maps can be generated in advance. If the accuracy of traditional saliency maps don't meet the requirements, please give comparison experiments on it.
2. Cannot understand why authors call this a conditional GAN, logD(IS,IF) actually equals logD(IS), no auxiliary label y is introduced, i.e. the equation would be logD(IS|y).

**Special Issue:**

no

---

> ### Author Response · Authors · 2020-03-26
> **Response to Reviewer 1**
>
> We thank the reviewer for the time, the positive assessment and raising some very thoughtful points. We address those in the following point-by-point response.
>
> Weaknesses:
>
> Rev.1:” Since the network architecture is based on (inspired by) previous work (PatchGAN), I think the authors should add experiments to validate the performance with this baseline”
>
> Resp.1: The design of the proposed network architecture is based on the pix2pix [1] method. In their work, they illustrate that using PatchGAN as the discriminator can achieve better results, and they also give us the qualitative and quantitative comparison to validate the performance (Figure 6 and Table 2 in [1]) and shown that the 70 x 70 PatchGAN gives the best performance. In this work, we directly use 70x70 PatchGAN as the discriminator, and do not compare it with traditional discriminator or even the PatchGAN with different receptive fields. We will add this comparison experiments in the revised version.
> [1] Isola P, Zhu J, Zhou T, et al. Image-to-Image Translation with Conditional Adversarial Networks[C]. computer vision and pattern recognition, 2017: 5967-5976.
>
> Rev.2:” Ablation study should be added to validate the effectiveness of the proposed loss function.”
>
> Resp.2: Thanks for your good suggestion which will make my paper more convincing. In this work, our main original contribution of the method part is the introduction of the local saliency loss, however, the comparison of the network with and without saliency loss is lack in our paper. To better demonstrate the effectiveness of the proposed method and local saliency loss, the comparison of the network trained with and without local loss will be added in the experimental results part of the revised version.
>
> Questions To Address In The Rebuttal:
>
> Rev.1:” As the authors claim that "Most common used saliency maps are pixel-based methods, which are time-consuming and do not meet the requirements of our work", What do authors mean "do not meet the requirements"? Obviously, for training process, local saliency maps can be generated in advance. If the accuracy of traditional saliency maps don't meet the requirements, please give comparison experiments on it.”
>
> Resp.1: Here we say that “the pixel-based methods cannot meet our requirement” only means that these methods do not meet our requirements in terms of computational efficiency, not accuracy. The local saliency loss proposed in this paper needs to calculate the saliency map of the real FFA image and the generated FFA image, and then calculate the Euclidean distance between these two saliency maps to ensure the training of the network. Therefore, the saliency map cannot be generated advance, it should be calculated during the training process. Based on the above reason, the computational efficiency of the saliency map is very important.
>
> Rev.2:” Cannot understand why authors call this a conditional GAN, logD(IS,IF) actually equals logD(IS), no auxiliary label y is introduced, i.e. the equation would be logD(IS|y).”
>
> Resp.2: In the proposed model, as shown in formula (2), the structure fundus image Is is the condition, the input of the Generator is {Is, n} (where n is a random noise vector), and the output is the generated FFA image G (Is, n), so the goal of the Discriminator is to distinguish the generated FFA image and the real image under the condition of structure fundus image (Is). Such a design can ensure the accurate generation of the image structure information.

---

### Official Review · AnonReviewer2 · 2020-03-13
**Application is interesting but has limited novelty, and better explanation of practical contribution is needed**

**Rating:** 3
**Confidence:** 4
**Recommendation:** Poster

**Summary:**

The paper applies conditional GAN to translate images from fundus images domain into fundus fluorescence angiography (FFA) images domain. A local saliency loss is proposed to facilitate the learning of small-vessel and fluorescein leakage features. The method has been validated with private dataset and the publicly available Isfahan MISP dataset.

**Strengths:**

1. The whole method is clearly described and details are explained to show the universality of the proposed method,.
2. The application itself is interesting and experiments have been performed on clinical datasets.

**Weaknesses:**

1. The motivation of image translation from fundus images to FFA images is not sufficiently explained. Compared with fundus images, can FFA images provide other more valuable information for physicians?
2. The validation of proposed method is limited. The authors should better validate the proposed local saliency loss to show its effectiveness, since the improved performance may come from the increased complexity of the model.
3. The comparison with other similar deep learning frameworks should be included to show the superiority of the proposed network.

**Detailed Comments:**

1. The process for the aligned image pairs is difficult to understand, especially for the image displayed in a circular region of interest (ROI). Better illustration of the whole process should be given for better understanding.
2. Ablation Study is needed for validation of the local saliency loss.


**Justification Of Rating:**

This paper presents a deep learning framework for transforming one image modality to the other using conditional GAN. The whole framework is in general well-presented. The topic seems interesting but still needs more evidences to show the motivation for doing this. The methodology contribution should be better validated in the experimental parts.

**Paper Type:**

validation/application paper

**Questions To Address In The Rebuttal:**

1. The author mentions in paper that “FFA is the only method to detect leakage”. It may indicate that fundus images does not contain any leakage information. Since the task of this paper is to generate FFA images with three common types of fluorescein leakage from fundus images. If the fundus images does not have leakage information, how is it possible to generate the fluorescein leakage?
2. Please provides more details to show that the generated FFA image is reliable and valuable for physicians?


**Special Issue:**

no

---

> ### Author Response · Authors · 2020-03-26
> **Response to Reviewer 2**
>
> We would like to thank the reviewer for taking the time to review this submission and the positive assessment. Please find below point-by-point responses to the reviewer’s comments.
>
> Because of the limit space, we cannot response the weakness part point-by-point, but we will add a more detailed description in the introduction part of the revised version to illustrate the motivation of our work and also add the performance comparison of the proposed model trained with and without local saliency loss to evaluate the effectiveness of the local saliency loss.
>
> Questions To Address In The Rebuttal:
>
> Rev.1:” The author mentions in paper that “FFA is the only method to detect leakage”. It may indicate that fundus images does not contain any leakage information. Since the task of this paper is to generate FFA images with three common types of fluorescein leakage from fundus images. If the fundus images does not have leakage information, how is it possible to generate the fluorescein leakage?”
>
> Resp.1: In our original paper, we mentioned that “FFA is the only method to detect leakage”, this does not mean the fundus image does not contain any leakage information. This means the FFA image and fundus image both contain the lesions’ information but in different morphology. For FFA imaging, because of the fluorescent dyes, it can make certain lesions, even not obvious in fundus image appear in the form of fluorescent leakage, which can help better detect and diagnose. And the FFA image synthetic method according to the specific lesion information or the potential information in the fundus image to generate the corresponding FFA image.
>
> Rev.2:” Please provides more details to show that the generated FFA image is reliable and valuable for physicians?”
>
> Resp.2: Until now, we can only compare the generated FFA image with the real FFA image in the generation of the whole structure, pathological structure and morphology, to illustrate the potential value of the FFA synthetic methods in medical applications. We do not yet have a better and more acceptable approach to verify the reliability and medical value of the method. But to find measurement to evaluate the reliable and valuable of the FFA image generation method is the problem we’ve been thinking about and studying, maybe it will be solved in our future work.
>
> Detailed Comments:
>
> Rev.1:” The process for the aligned image pairs is difficult to understand, especially for the image displayed in a circular region of interest (ROI). Better illustration of the whole process should be given for better understanding.”
>
> Resp.1: For aligned images, in the aligned part, the two images are overlapped. Since our method needs paired images, we only need to extract the patches (512x512) from the aligned part in the structure fundus and FFA image pairs. The original fundus and FFA images in our dataset displayed in two forms. One is displayed as a square image (Fig.3), and the other is displayed in a circular ROI (Fig.1). For the second form, the extracted patch pairs may contain both the image and the black circular edges, or one only contains the retinal part and the other contains both retinal and black circular edges. Using these kinds of image pairs as training data could affect generation performance and make network is not robust to the different image displayed forms. Thus, we design a mask with a circle to obtain the image pairs displayed as the second form.
> The description of this part in the original paper is not very clear, we will revise the description in our revised version.
>
>
> Rev.2:” Ablation Study is needed for validation of the local saliency loss.”
>
> Resp.2: In this work, our main original contribution of the method part is the introduction of the local saliency loss, but we lack the comparison of the network with and without saliency loss. To better demonstrate the effectiveness of the proposed method, the performance comparison of the network trained with and without local saliency loss will be added in the revised version.

---

### Official Review · AnonReviewer4 · 2020-03-18
**Well-designed and well-written paper with no substantial contribution in technical issues, but good experimental results in a new clinical application**

**Rating:** 3
**Confidence:** 4
**Recommendation:** Oral, Poster

**Summary:**

This paper proposes to generate artificial fluorescent images, both Fundus Fluorescence Angiography and Fluorescein Leakage, using GAN networks. Although the authors use some traditional concepts, such as saliency map and adversarial network, they developed new approaches to generate background and foreground images and some modifications on the loss function. Some experiments with HRA and MISP dataset were developed with acceptable PSNR and SSIM values.

**Strengths:**

This paper is clear and well-written. Methodology explains properly the way to calculate the saliency map, the conditional adversarial network and the loss function. Maybe, the most relevant contribution is a successful adaptation of previous knowledge and techniques to a specific clinical application. So, needed changes of previous architecture, loss function and other technical issues to fundus retinal images are properly developed and tested in this work.

**Weaknesses:**

Unfortunately, Isfahan MISP dataset is not available at the moment I try to get access. Furthermore, the HRA dataset has not been published by the authors (as far as I know). So, this issue makes difficult future fair comparison of results.

Although PSNR and SSIM are two well-known quantitative metrics of quality I would suggest the authors for future works to consider other metrics (see https://arxiv.org/abs/1802.03446). Maybe, these quality measures are not the most standard metrics to evaluate generated images using GAN networks.

**Detailed Comments:**

As far as I know, the proposed architecture was previously used with residual blocks [1] or skip connections or dense connections [2][3]. All of them in other fields of interest. Here, this work uses different number and size of blocks or activation functions. Additionally, the proposed strategy of use two loss functions is previously analysed [4]. However, I do not completely understand these concepts introduced by the authors: global and local loss functions. Is local loss referred to the saliency map of each image? And is global loss referred to a general value of the whole training set? Although I carefully read the description and equations included in the section 2.4 I would thank some additional explanations, e.g. is L1 loss the same as Lpixel?
As far as I can understand, the only original contribution on this issue (loss functions) is the local saliency loss, am I right?

[1] https://www.researchgate.net/figure/Summary-of-the-GAN-architecture-In-the-bottom-left-we-show-the-pipeline-We-detail-both_fig1_328015657

[2] https://www.researchgate.net/figure/Architecture-of-the-generative-adversarial-network-GAN-based-method-with-corresponding_fig1_335862244

[3] https://www.researchgate.net/figure/The-proposed-condition-generative-adversarial-network-cGAN-for-FPM-video_fig2_327914814

[4] https://developers.google.com/machine-learning/gan/loss#minimax-loss

Some minor, grammar or spelling improvements:

-	Introduction: the sentence “potential effects including death” maybe it is quite alarming. Even this old reference estimated death to occur in 1:221,781 (Yannuzzi LA, Rohrer, MA, Tindel LJ, et al. Fluorescein angiography complication survey. Ophthalmology 1986;93:611-7)
-	Page 3. There are three common types of leakage that can be…
-	Page 5. Secondly, a Gaussian filtering…
-	Page 5. Ib and If (background and foreground images) are named later in the figure and the rest of paper such as IB (uppercase b) and Ifiltered. I suggest unifying the terminology for the whole paper.
-	Page 9. AS shown in Figure 5… it is a very large sentence. I strongly suggest splitting in several ones.
-	Page 12. generation instead of genration (several times)
-	Page 12. Puncate focal. I suppose the authors refers to ‘punctate focal’


**Justification Of Rating:**

Although technical novelty of this paper is minimal from my point of view, the original contributions (adaptation of architecture and local saliency loss) are interesting ways to solve this specific problem. Maybe, the most important point for my final rating is the experimental results that shows a good behavior of the adversarial network. However, I do not consider such as strong accept, mainly because of the used metrics are only focused on quality image criteria.

**Paper Type:**

both

**Questions To Address In The Rebuttal:**

Please, some additional details should be included about local and global loss. I include some comments in the next section related with this point. As far as I understand, it is not completely clear for readers how are these hyperparameters calculated. Do you think it is possible find an understandable meaning for these hyperparameter values?

**Special Issue:**

no

---

> ### Author Response · Authors · 2020-03-26
> **Response to Reviewer 4**
>
> We thank the reviewer for their time and positive assessment of our work. Please find below point-by-point responses to the reviewer’s comments.
>
> Questions To Adress In The Rebuttal:
>
> Rev.1:” Please, some additional details should be included about local and global loss. As far as I understand, it is not completely clear for readers how are these hyperparameters calculated. Do you think it is possible find an understandable meaning for these hyperparameter values?”
>
> Resp.1: In this work, we formulate the loss function as a combination of global loss and local; where global loss is a combination of adversarial, pixel-space, and feature-space loss; and local loss is based on the local saliency loss. The global loss is used to ensure the generated FFA image similar to the real FFA image and with good visual effect. The local loss can make the network more concentrate on generating more accurate fluorescein leakage structures and vascular structures. And the hyperparameters (weights) of these losses are used to control the effect of these losses.
> The selection of hyperparameters of pixel-space loss and feature-space loss is based on the papers [1-2]. Since the l1 distance between real and generated image is a very small value at the start of the training process, to avoid this small loss affect the training, we give it a weight equals to 100. Since the perceptual distance between real and generated image is a relatively larger value, to ensure the network can take into account the image intensity information and perceptual information reconstruction during the training process, we give it a weight with 0.001. Moreover, with a large weight of perceptual loss will lead to the loss of image intensity [2]. For local saliency loss, the Euclidean distance between the generated image and real image local saliency map is a relatively suitable value, give a large weight of local saliency map will make the network ignore the reconstruction of the background areas, otherwise, will make the network pay insufficient attention to the lesion area. Thus, we finally give local saliency loss a parameter with 1. The parameters of these losses were all determined by experiments, which are found to be the suitable parameters to achieve better performance.
> [1] "Image-to-Image Translation with Conditional Adversarial Networks", CVPR, 2017.
> [2] "DeblurGAN: Blind Motion Deblurring Using Conditional Adversarial Networks", CVPR , 2018.
>
> Detailed Comments:
>
> Rev.1:” However, I do not completely understand these concepts introduced by the authors: global and local loss functions. Is local loss referred to the saliency map of each image? And is global loss referred to a general value of the whole training set? Although I carefully read the description and equations included in the section 2.4 I would thank some additional explanations, e.g. is L1 loss the same as Lpixel?”
>
> Resp.1: In this work, to ensure the generated FFA image similar to the real FFA image, we designed a loss, which is a combination of global loss and local loss. Here, the “global” and “local” is defined in the image level, that is “global loss” concentrate more on the reconstruction of the whole information of the image, and “local loss” concentrate more on local information reconstruction, and the local information in this work refers to the information of lesion areas and vascular structures.
> Since the saliency of FFA image can reflect the vascular structures, optic disc, and lesions, we designed a local saliency loss as the local loss to ensure the accurate generation of these pathological structures. And the local saliency loss is defined as the Euclidean distance between the generated image and the real image local saliency map.
> In addition, in this work, we use the L1 loss as the pixel-space loss to ensure the accurate reconstruction of pixel-level information, since L2 loss usually leads to blurry artifacts [1-2].
> [1] “Deep multi-scale video prediction beyond mean square error,” CVPR, 2015
> [2] “Context encoders: Feature learning by inpainting,” CVPR, 2016
>
> Rev.2:”As far as I can understand, the only original contribution on this issue (loss functions) is the local saliency loss, am I right? ”
>
> Resp.2: Indeed, in the method part, our main original contribution is the local saliency loss. Since the existing FFA image synthetic methods cannot accurately generate the pathological structures, the introduction of the local saliency map and loss can make the network concentrate more on the pathological structures and give better generation performance. I know that the original paper lack of the performance comparison of the model with and without local saliency loss, we will add this comparison in the revised version to demonstrate the effectiveness of the local saliency loss.
>
> Rev.3:” Some minor, grammar or spelling improvements:
>
> Resp.3: These embarrassing grammar and spelling mistakes will be revised in the revised paper.

---

### Meta-Review · Area_Chair1 · 2020-04-10
**MetaReview of Paper333 by AreaChair1**

**Rating:** 3
**Recommendation For Accepted Papers:** Poster

**Metareview:**

This paper presents a method to generate  Fundus Fluorescence Angiography  from conventional fundus imaging. The work is based on a GAN framework modifying the loss function to include both local and global terms. All reviewers are very consistent on the clarity of the work and the thoroughness of the evaluation while also indicating that there is limited novelty on the methodology. I recommend this paper be accepted if possible. The score would be around 3.3.

**Paper Type:**

validation/application paper

**Special Issue:**

no

---

### Decision · Program_Chairs · 2020-04-11

Accept